# Plant Secondary Metabolites in the Battle of Drugs and Drug-Resistant Bacteria: New Heroes or Worse Clones of Antibiotics?

**DOI:** 10.3390/antibiotics9040170

**Published:** 2020-04-10

**Authors:** Cyrill L. Gorlenko, Herman Yu. Kiselev, Elena V. Budanova, Andrey A. Zamyatnin, Larisa N. Ikryannikova

**Affiliations:** 1Institute of Molecular Medicine, Sechenov First Moscow State Medical University, 119991 Moscow, Russia; mr.gorlenko@mail.ru (C.L.G.); kiselev32rus@gmail.com (H.Y.K.); e.v.budanova@mail.ru (E.V.B.); 2Belozersky Institute of Physico-Chemical Biology, Lomonosov Moscow State University, 119992 Moscow, Russia

**Keywords:** plant secondary metabolites, bacterial drug resistance, pathogens, antimicrobial activity

## Abstract

Infectious diseases that are caused by bacteria are an important cause of mortality and morbidity in all regions of the world. Bacterial drug resistance has grown in the last decades, but the rate of discovery of new antibiotics has steadily decreased. Therefore, the search for new effective antibacterial agents has become a top priority. The plant kingdom seems to be a deep well for searching for novel antimicrobial agents. This is due to the many attractive features of plants: they are readily available and cheap, extracts or compounds from plant sources often demonstrate high-level activity against pathogens, and they rarely have severe side effects. The huge variety of plant-derived compounds provides very diverse chemical structures that may supply both the novel mechanisms of antimicrobial action and provide us with new targets within the bacterial cell. In addition, the rapid development of modern biotechnologies opens up the way for obtaining bioactive compounds in environmentally friendly and low-toxic conditions. In this short review, we ask the question: do antibacterial agents derived from plants have a chance to become a panacea against infectious diseases in the “post-antibiotics era”.

## 1. Introduction

Since ancient times, people have used various plants and their derivatives for medical purposes, including the treatment of infectious diseases. Perhaps the most striking example is quinine, an alkaloid from the bark of the cinchona tree, which was very widely used not only to treat malaria, but also to treat other infectious diseases, like pneumonia, typhoid fever, and even ordinary nasopharyngeal infections [1]. Cinnamon is another wonderful example. This substance is primarily known as a seasoning for food, but in ancient Chinese or Indian medicine, cinnamon was used as a multipurpose remedy. Its main biologically active agent, cinnamaldehyde, proved to be an efficient antimicrobial agent [2,3]. There are recipes for folk or traditional healing practices that use the biological activity of various substances that are derived from plants to treat different diseases, including those that are caused by bacteria. Many of these traditional treatments are still widely used today. Moreover, some commercially established drugs used in modern medicine had an initial crude form in folk medicine [4,5].

The beneficial remedial effects of plant materials are mainly due to the mixture of substances called secondary metabolites of plants (SMoPs). SMoPs is a diverse biochemical group of substances produced by the plant cell through secondary metabolic pathways that are derived from the primary metabolic pathways. In contrast to the primary metabolites involved in the main metabolic pathways vital for survival, SMoPs are not essential for growth and life, but they play important roles in interspecies competition and defence, including protecting plants against herbivores and microbes [6,7,8,9,10,11].

Presently, about two-hundred thousand different SMoPs have been isolated and identified [12]. They can be classified based on their chemical structures and/or biosynthesis pathways [6,8,11,13,14]. A simple classification includes three main groups: terpenoids (polymeric isoprene derivatives and biosynthesized from acetate via the mevalonic acid pathway), phenolics (biosynthesized from shikimate pathways, containing one or more hydroxylated aromatic ring), and alkaloids (non-protein nitrogen-containing compounds, biosynthesized from amino acids, such as tyrosine). Together, these groups make up about ninety percent of all SMoPs [14]. The minor groups include saponins, lipids, essential oils, carbohydrates, ketones and others [6,15].

Many SMoPs are widely used in the pharmaceutical and food industries, in perfume, agrochemicals, and cosmetics production [4,11,12,14,16,17]. In the current review, we focus on the possibility of using SMoPs as antibacterial agents against important human pathogens that could act independently or enhance the action of the conventional antibiotics. This topic is extremely relevant today with the emergence and spread of completely drug-resistant strains of microorganisms. Many questions remain despite the fact that the antibacterial properties of secondary plant metabolites have been studied for a long time [15,18,19,20,21,22,23,24,25,26,27,28]. First, in the huge sea of substances that are produced by plants, some are already known about, but still insufficiently studied and some are not yet discovered. There is a good chance that we will find some novel compounds that demonstrate antibacterial activity. Why should we expect this? The reason is that many of the secondary metabolites are used by plants themselves as a defence mechanism against pathogens. Therefore, they have the capacity to partially or completely inhibit the proliferation of some microorganisms. This type of action can also be expected to extend to animal and human pathogens [29].

Modern science, using its newest approaches of high-performance and large-scale screening, offers new ways to detect novel metabolites produced even by well-known plants. Another clear advance is the possibility of producing large quantities of bioactive substances while using modern gene engineering approaches or chemical synthesis methods. In this short review, we propose the following question: taking into account the current trends in science, are there the prerequisites for the transition from the era of antibiotics of microbial origin and their derivatives to the era of "plant antimicrobials"?

## 2. SMoPs Discovering and Manufacturing: New Times Provide the Opportunities

For hundreds or even thousands of years, people have used a fairly limited set of methods for obtaining SMoPs, including extraction, extrusion, distillation, infusion and fermentation, enfleurage, and concentration [30,31,32,33]. Over time, these methods, without changing in essence, became more advanced and productive enabling us to obtain more refined substances. Nevertheless, the natural extracts are incredibly complicated and they are usually composed of hundreds to thousands of metabolites, of which many are represented in ultra-small quantities. We still cannot claim that all or even most of the plant-derived metabolites have been identified and characterized. In addition, the bioactivity of natural extracts can be represented by synergism between several compounds. Therefore, the key, and often the most challenging aspect of research, is not only to quantitatively measure the bioactivity of any plant extract, but also to connect a particular chemical structure(s) with a particular clinical effect [4].

The development of science in the twenty-first century offers new approaches, enabling us not only to discover and identify even the ultra-small quantities of the compounds produced by plants, but also to produce them in sufficient quantities to reliably characterize their properties, such as bioactivity. Metabolomics has emerged in recent years as an indispensable tool for the analysis of thousands of metabolites from crude natural extracts, leading to a paradigm shift in natural drug products [34,35,36,37].

The pathway from a plant to the end product—drugs, flavours, colourants, sweeteners, antioxidants, or nutraceuticals—is still quite long. Currently, the search for effective approaches towards producing plant substances continues in two extensive ways. The first one is the development of advanced extraction techniques to obtain biologically active compounds from fresh plant material or remaining waste [38]. Modern complex extraction techniques have gradually replaced conventional ones that demand a long processing time, high solvent and energy consumption, and large quantities of raw material. These conventional methods rarely produce a significant quantity of the active compound. The innovative technologies that are based on supercritical fluid extraction [39,40], microwave or ultrasound power [41], or membrane separation technology [42] can significantly help to overcome the disadvantages of classical methods. Other non-conventional techniques, such as electrotechnologies (high voltage electric discharge or pulsed electric field), are promising tools for the isolation of bioactive compounds from plant material [38,43] (Figure 1). The second way involves biotechnological techniques that lead to the production of plants with increased levels of fine chemicals, new compounds with potential biological activity. Normally, many secondary metabolites are present in plants at very low concentrations, which eliminates the possibility of using wild-growing plants for manufacturing these important products. Occasionally, the structural and stereochemical complexity of specialized metabolites hinders most attempts to access these compounds using chemical synthesis. Currently, plant cells, tissues, and organs are artificially grown in shaken flasks and bioreactors (the so-called “green cell factories” concept). These in vitro plant technologies are considered to be cost-effective and eco-friendly alternatives to the wild harvest of biomass for the mass production of plant-derived molecules [38,44,45]. In addition, the metabolic bioprocess is fully independent of any seasonal and geographical conditions [4,46]. Moreover, genetic modifications can be readily applied to increase output, reduce toxin levels, and increase the uniformity and predictability of the desired compounds. Furthermore, such technologies appear to be the only economically feasible way of producing some high value metabolites from rare and threatened plants [38,44].

The next important step is the isolation, purification, and detailed characterization of active metabolites from crude plant extracts. Recent developments in analytical chemistry platforms, such as mass spectrometry supplied with gas/liquid chromatography or capillary electrophoresis, and nuclear magnetic resonance (NMR) spectroscopy, have led to highly efficient tools for metabolome analysis, allowing for the detailed characterization and ultimately the structural elucidation of these agents [34,47,48]. At the final stage, the biological activity, i.e., the effects in cell lines, animal models or human volunteers, is screened for assessing the pharmacological potential of the candidate compounds. One more important area is the structural modification of natural plant compounds that possess bioactivity, in order to develop novel compounds with more specific properties. Although naturally active substances are good material for the development of new drugs, most of them suffer from various deficiencies or shortcomings, such as complex structures, poor stability, or solubility. Chemical modification of plant compounds increases the activity or selectivity of antibiotics, improving their stability or physico-chemical properties [49,50].

## 3. Antimicrobial Properties of SMoPs: The Reserve Players Against the Life-Threatening Pathogens

In general, a plant’s secondary products may exert their common beneficial medicinal actions on humans in indirect ways, such as by resembling endogenous metabolites, ligands, hormones, signalling molecules, or neurotransmitters [5]. However, here we discuss the direct impact of these compounds on microorganisms, namely the inhibition of the microbial growth when exposed to any SMoPs.

There is a large number of reports on antimicrobial activity of SMoPs. In Table 1, the examples are summarized that illustrate the effects of different SMoPs on a number of important human pathogens. 

These pathogens were determined by WHO in 2017 as the most life-threatening bacteria due to their rapidly developing resistance to drugs; this list includes, among others, those pathogens that the WHO has assigned a critical priority I and II—the so-called ESKAPE group (an acronym for *Enterococcus faecium, Staphylococcus aureus, Klebsiella pneumoniae, Acinetobacter baumannii, Pseudomonas aeruginosa,* and *Enterobacter* spp.) [83]. We added *Mycobacterium tuberculosis* to this list, which is one of the most dangerous pathogens since ancient times, which has not yet been eradicated despite the efforts of the medical and the scientific community. As can be seen, many substances that have been extracted from plants have been found to demonstrate bactericidal or bacteriostatic activity against the microorganisms listed. Certain microorganisms are now of particular concern. For example, *Staphylococcus aureus*, one of the most dangerous human pathogens, causing a wide range of infections from mild skin diseases to life-threatening endocarditis [23,84]. The greatest problem is the methicillin-resistant (MRSA) *S. aureus* strain, which is the fastest-evolving pathogen in the last decade and one of the most common causes of multidrug-resistant infections with significant morbidity and mortality, especially in developing countries [85]. After identifying methicillin-resistant strains, vancomycin and quinolones antibiotics have been used as alternative drugs of choice in staphylococcal infections therapy [86]. However, their effectiveness is declining, and scientists are looking for the new ways to counter the effects of MRSA antibiotic resistance [87]. In the last decade, many reports have affirmed the strong antimicrobial action of some SMoPs (alone or in combination with antibiotics) against *S. aureus* strains, including MRSA. Some of these may provide a sustainable solution to drug-resistant microbes (Table 1). The search continues for a drug against *M. tuberculosis*, the pathogen causing tuberculosis. Tuberculosis has the second highest fatality rate after HIV [23,88,89,90]. This extremely notorious and infectious disease causes thousands of deaths per year worldwide. Since the 1990s, the incidence and mortality from tuberculosis has dramatically increased. One of the reasons for the difficulty in treating this disease is the widespread multidrug-resistance (MDR), extensive drug-resistant strains (XDR), and total drug-resistant strains (TDR), which are non-susceptible to either the first-line drugs (especially rifampicin and isoniazid) or the second-line drugs (fluoroquinolones, aminoglycosides, etc.) [89]. It was demonstrated that plant-derived compounds could have significant anti-mycobacterial activity against *M. tuberculosis* (see Table 1 and review [23]), confirming that phytochemicals could be useful as ancillary solutions to control this infection.

Returning to Table 1, note that, despite the demonstration of an apparent antibacterial effect, generally, plant-derived metabolites seem to be inferior in efficiency when compared to modern high-effective antibiotics of microbial origin whose minimal inhibitory concentration is calculated in tenths and hundredths of micrograms per millilitre. However, the development of resistance and cross-resistance is a serious drawback to the use of current antimicrobials. Therefore, it is possible that it will plant phytochemicals that are assigned the role of true lifesavers against life-threatening infections in the future. Nevertheless, the question concerning the possible development of bacterial resistance to the plant-derived metabolites still remains. This issue is discussed below.

## 4. Mechanisms of SMoPs Antimicrobial Action: A New Weapon Against the Old Targets?

The mode of action of plant secondary metabolites relies on their chemical structure and properties. SMoPs can affect the microbial cell in several different ways. These include the disruption of cytoplasmic membrane function and structure (including the efflux system), interaction with the membrane proteins (ATPases and others), interruption of DNA/RNA synthesis and function, destabilization of the proton motive force with leakage of ions, prevention of enzyme synthesis, induction of coagulation of cytoplasmic constituents, and interruption of normal cell communication (quorum sensing) [15,21,91]. For many classes of SMoPs, these mechanisms have been well studied. Thus, we know that the alkaloids possess the ability to intercalate with DNA, thereby disrupting transcription and replication, and can also inhibit cell division, thereby resulting in cell death [26,92]. For example, berberine, which is a well-known phytochemical of the alkaloid group from *Berberis spp*., can severely damage the structure of bacterial cell membranes and inhibit the synthesis of proteins and DNA under interaction with *Streptococcus agalactiae*. This was shown by means of TEM and SDS-PAGE of membrane proteins [93]. The potential and character of the interaction of this cationic molecule with a polyanionic double-strand or single-strand DNA floating in solution or immobilized on the glassy carbon electrode was elucidated using electrochemical techniques by Tian et al [94]. The antimicrobial activity of flavonoids results from their action on the microbial cell membrane; they interact with membrane proteins that are present on bacterial cell walls increasing the permeability of the membrane and disrupting it [92,95,96]. The antimicrobial action of terpenes and terpenoids as well as essential oils is also mostly attributed to their ability to interact with and destroy microbial membranes [92,97]. Carvacrol and thymol, the two most studied monoterpenes obtained from *Thymus vulgaris*, have the ability to integrate into bacterial cell membranes due to their hydrophobic nature, causing disruption and disturbance to normal membrane function leading to increased permeability of ATP and an increased release of other cellular components [98,99]. Khan, who used scanning electron microscopy to demonstrate an interaction of carvacrol with the lipid bilayer of *Escherichia coli,* confirmed this mechanism [100]. The major targets of plant-derived quinones in the microbial cell are assumed to be the surface-exposed adhesin proteins, cell wall polypeptides, and membrane-bound enzymes [92,95,101]. The effect of antimicrobial efficacy of polyphenols and tannins is possibly due to an inactivation of cell envelope transport proteins, enzyme inhibition, or disruption of membranes [92,95,102]. (Figure 2) 

We refer to some excellent reviews for a more detailed consideration of these mechanisms [15,21,91]. Here, we would like to address the question of the resistance of microorganisms to drugs and ask whether the resistance that is currently seen to conventional antibiotics of microbial origin could also arise in relation to the antibacterial agents that are produced from plants. We expect that this is highly likely. Such an assumption is justified since the targets of plant-derived metabolites in the bacterial cell are, in fact, the same as the targets of routinely used antibiotics, i.e., the cell membrane, or the growing DNA chain, or intracellular enzymes, and so on. Some examples of “herbal drug resistance” are given in the excellent review by Vadhana [106], which predicts that the number of reports of bacterial resistance to herbal antimicrobials will increase. It has been reported that some microorganisms, including multidrug-resistant strains of *E. coli*, *Klebsiella pneumoniae*, *S. aureus*, *Enterococcus faecalis*, *Pseudomonas aeruginosa,* and *Salmonella typhimurium*, can demonstrate a non-susceptibility to some components of the herbal medicines [28], perhaps having natural resistance to them. In other investigations, cases were described where drug-resistant or MDR strains (including strains listed by WHO) were directly isolated from herbal products, such as garlic, onion, ginger, rosemary, or mustard powders or liquids, all of which were assumed to have strong antibacterial properties [107,108].

Details of the resistance mechanisms of microorganisms against these compounds are not yet clear. It is often stated that bacteria do not develop resistance to herbal medicines, or at least the level of resistance is still low [91]. However, taking the fact that many plant derivatives are actively used now for food, medicine, or cosmetics into account, it can be assumed that the spread of "herbal drug resistant" strains, as in the case of conventional antibiotics of microbial origin, is only a matter of time. However, the bioactivity of plant extracts is composed of bioactivities of many SMoPs, so the resulting medicinal effect might be due to the combined or synergistic actions of various phytoactive components directed at multiple targets in the bacterial cell. Thus, we might expect that the development of bacterial resistance to such synergistic combinations might be much slower than that for single chemical compounds [17,92].

## 5. Overcoming the Bacterial Drug Resistance

It is well-known that the bacterial cell can inactivate drugs by means of a number of mechanisms. These mechanisms include both the “classical” ways, such as a modification of drug targets, pumping out the damaging agent from the cell (efflux) or enzymatic inactivation of the drug, and “non-classical” ways, for example, the mechanistic protection provided by biofilm formation. As to the ability of plant-derived compounds to overcome or help overcome the microbial resistance, data have been collected for addressing some special ways to prevent bacterial cells escaping eradication.

### 5.1. Plugging the Efflux Pumps

Efflux pumps are cell systems that are crucial for stress-adaptations, virulence, and pathogenicity. They are considered to be vital components for the development of antimicrobial resistance in pathogens, enabling toxic substances to be actively pumped out of the cell. Therefore, the compounds that can inhibit efflux pump activity are extremely important in overcoming drug resistance. There are numerous reports confirming that SMoPs are able to effectively inhibit these powerful pumps within the bacterial cell. Many medicinal plants with antimicrobial potential have been reported to comprise efflux pump inhibitors, among them catechol, piperine, quercetin, resveratrol, and many others (see Table 1) [109,110,111]. Although the exact mechanisms remain to be clarified, there are some predictions of how plant-derived efflux inhibitors work. SMoPs can occlude the canal that is involved in the process of evacuation of substrate. For instance, totarol, a diterpene from *Podocarpus totara*, acts as a concurrent inhibitor of NorA-pump in *S. aureus* [112,113,114] (Figure 3). Additionally, polyphenolic molecules can bind directly to structural proteins of the efflux pump canal that can cause conformational changes and stop the elimination of the substance [115]. Often, SMoPs act on efflux pumps as synergists for the antibacterials when used in combination. Thus, alkaloid reserpine from *Rauvolfia vomitoria* inhibits NorA efflux pump activity in *S. aureus* in combination with fluoroquinolones [116], while ferruginol from *Sequoia sempervirens* blocks etidium bromide efflux in combination with norfloxacin in this pathogen [117].

### 5.2. Attenuating the Bacterial Virulence

In some cases, the plant extracts may exert their antimicrobial activity by affecting key events in the pathogenic process. Qiu et al. reported that a treatment with subinhibitory concentrations of thymol or eugenol decreased the production of α-haemolysin and staphylococcal enterotoxins A and B in both methicillin-sensitive and methicillin-resistant *S. aureus* isolates [70,118]. Similar results have been obtained that demonstrate reduced S. aureus haemolysis activity and a decrease or even inhibition of the production of staphylococcal α-haemolysin after the treatment of allicin [119], alkaloid capsaicin [120], flavonoids farrerol [121], or epicatechin gallate [122]. Allicin, the major biologically active component of garlic, was shown to effectively neutralize the toxin pneumolysin—a main virulence factor that is produced by *S. pneumoniae* [123]. In the same way, coumarin derivative esculetin repressed Shiga-like toxin gene stx2 in *E. coli* and attenuated its virulence in vivo [124].

### 5.3. Disrupting the Biofilms

Bacterial biofilms are one feature of bacterial life that helps them to survive in unfavourable environmental conditions, including drug pressure. Bacterial biofilms are the complex structures, representing a community of microorganisms that are attached to any surface and surrounded by a biopolymer matrix. These structures provide a complex regulation mechanism that is based on intercellular communication. The unique ability to survive within the biofilm is due a number of reasons: the presence of persisting cells (persisters) possessing delayed metabolism, the filtering capacity of the biopolymer matrix hindering the diffusion of drugs, and a genetic "cooperation" and mutual assistance of bacterial cells allowing for microbe communities to manage their life resources and flexibly respond to the changeable environmental conditions. Biofilms that are typically the cause of chronic, nosocomial, and medical device-related infections are the great problem in the clinic due to their high tolerance to antibiotics.

Many studies have been devoted to the question of the effect of SMoPs on the biofilm structure, and many plant extracts have been identified that control biofilm formation and growth in major human pathogens. Appendix A summarizes some examples. Phenylpropanoids, such as eugenol and cinnamaldehyde, terpenoids (thymol and carvacrol), betulinic and ursolic acids, alkaloids, such as berberine, indole, or chelerythrine, and other plant-derived compounds were found to exhibit marked anti-biofilm activity against *P. aeruginosa* [125,126,127,128,129,130,131], *K. pneumoniae* [132,133,134], staphylococcal biofilms [135,136,137,138,139,140], both affecting the pre-formed biofilms and preventing the formation of new ones. The anti-biofilm actions of SMoPs are believed to be realized in different ways, such as the disruption of intercellular communication, disturbance in cell-to-cell coaggregation, inhibition of cell mobility, inactivation of bacterial adhesins, or stimulation of bacteria dispersal [56,141]. 

### 5.4. Blocking the Interbacterial Communication

Quorum sensing (QS) is a complex system regulating cell-to-cell communication in the microbial population, and the ability to interfere with QS thereby interrupting bacterial communication, would open up new therapeutic prospects. A number of plant extracts and natural compounds reducing QS-mediated gene expression in *P. aeruginosa* have been identified, including the organosulfur ajoene from garlic or isothiocyanate iberin from horseradish [142,143], sulforaphane (*Brassica oleracea*) [144], flavonoids naringenin and taxifolin [145], and quercetin [146], extracts from the flowers of *Chamaemelum nobile* [147] or *Kalanchoe blossfeldiana* leaves [148]. It was shown that caffeine demonstrates anti-QS properties against *P. aeruginosa* inhibiting the production of AHL (N-acyl homoserine lactone) signalling molecules [149,150]. Similar observations have also been published for other pathogens [141,151,152,153,154]. Two major mechanisms of QS-inhibition by SMoPs are assumed: the first one consists of the down-regulation of QS genes, resulting in a lower expression of signal molecules, while the second mechanism includes interaction between SMoPs and a QS signal molecule resulting in the inactivation of quorum mediators and a decreasing intensity and effectiveness of cell-to-cell interaction [141,151].

## 6. Safety in Numbers: Synergism in "Metabolite-Metabolite" or "Metabolite-Drug" Systems

Each phytochemical compound demonstrating a high level of bactericidal activity has the potential to stimulate the development of microbial drug resistance, as assumed above. However, a crude extract consists of multiple components, each of them being able to act at different sites of the microbial cell thereby contributing to the overall activity of the extract [92,155]. In fact, it has been repeatedly demonstrated that the process of isolation of some phytochemicals often leads to a loss or reduction in their activity. One good example is an investigation of the minimum inhibitory concentration (MIC) of oregano essential oil and two of its principal components—thymol and carvacrol—against *P. aeruginosa* and *S. aureus*. The additive antimicrobial effect of carvacrol and thymol, as well as the overall inhibition by essential oil, appeared to be much more than when any of the two components were used alone [156]. Moreover, the development of bacterial resistance to such combinations might be much slower than that to single chemical compounds.

Additionally, in conjunction with the routinely used antibiotics, SMoPs can demonstrate diverse combinatorial effects [157,158,159]. It is known that multidrug therapy of SMoPs with each other and/or antibiotics might have an insignificant, additive, synergistic, or antagonistic effect. The insignificant (or neutral) effect is observed when the overall antimicrobial effect of two compounds is the same. There is no visible profit in the use of such combinations. The additive effect occurs when the cumulative antimicrobial effect is a sum of the effects of individual compounds. The synergistic effect is observed when the antimicrobial activity of a combination of compounds is higher than the sum of the effects of individual compounds. Finally, the antagonism means that the activity of a combination of compounds is lower than the activities of individual compounds [22,27,160,161]. In Appendix A, some examples of synergistic combinations of SMoPs and antibiotics are given, illustrating the significant reduction of bactericidal concentrations in comparison with antibiotics that are given without SMoPs.

It is obvious that we need to reveal the synergistic combinations that result in a decrease in the minimal inhibitory concentration of standard antimicrobial drugs. In addition, knowledge of the molecular mechanisms of synergistic behaviours of plant compounds would help to develop new ways to overcome the rise of MDR pathogens, thus reducing the overuse of antibiotics and their side effects.

## 7. Conclusion and Future Perspectives

Keeping in mind the increasing worldwide resistance of dangerous bacterial pathogens to current antibiotics, the search for new effective antibacterial agents is now a task of top priority. In the last two decades it has become clear that overcoming antibiotic resistance by developing more powerful antibiotics on the basis of old principles and old chemical classes, can only lead to limited and temporary success and it will contribute to developing even greater resistance. In this regard, the plant kingdom appears to be a bottomless well of novel antimicrobial agents that is unlikely to be quickly exhausted. Plants are readily available and cheap; extracts or compounds from plant sources often demonstrate a broad spectrum of activity against pathogenic species, rarely have severe side effects, and often possess the immunomodulatory action in humans. The enormous variety of plant-derived natural compounds provides very diverse chemical structures that may supply both the novel mechanisms of antimicrobial action and new targets within the bacterial cell. In addition, the rapid development of modern biotechnologies opens up the way for obtaining bioactive compounds in an environmentally friendly and low-toxic way.

Obviously, each compound that is extracted from a plant is not ready to be instantly used in routine clinical practice. We need antibacterials with sufficiently low inhibitory concentrations, minimal toxicity, and ease bioavailability for efficient and safe use in humans. Current advances in bioscreening research, including the omics technologies, first of all metabolomics, will enable us to both catch and identify even very low-quantity active phytochemicals and clarify the specific molecular mechanisms underlying their effect(s) on bacterial targets. Another promising and essential field is the modification of the chemical structure of potentially useful compounds, to improve their antibacterial properties, and decrease their toxicity and side effects.

Research shows that some SMoPs possess high-levels of intrinsic antibacterial activity. However, it should keep in mind that, even in the case when a plant-derived substance reveals strong antibacterial effects, there is always the possibility that bacteria will appear to be non-susceptible or develop resistance to it. Therefore, a way to combine plant metabolites with conventional antibiotics might be the most profitable. Such combinations act at different target sites in bacterial cells and lead to high levels of efficacy, especially in suppressing the development of resistance. Currently, there is much evidence to suggest that combining antibiotics with SMoPs or plant-derived extracts results in improved pharmacological activity, at the same time minimizing the likelihood of dose-dependent toxicity that is mediated by synthetic chemicals. It is clear that a detailed understanding of the molecular mechanisms underlying the action of phytochemicals, or of those underlying phytochemical-antibiotic interactions, is required for developing a successful therapeutic approach. These mechanisms are likely to be the major subject of future research.

## Figures and Tables

**Figure 1 antibiotics-09-00170-f001:**
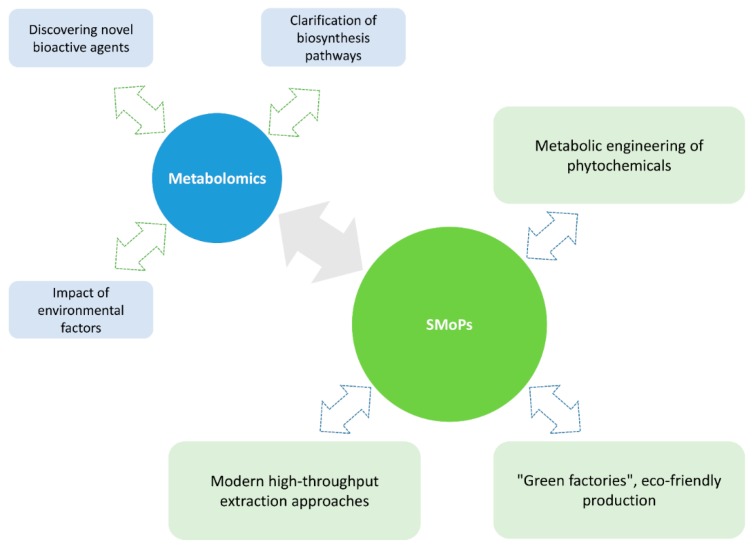
Herbal manufacture in the twenty first century.

**Figure 2 antibiotics-09-00170-f002:**
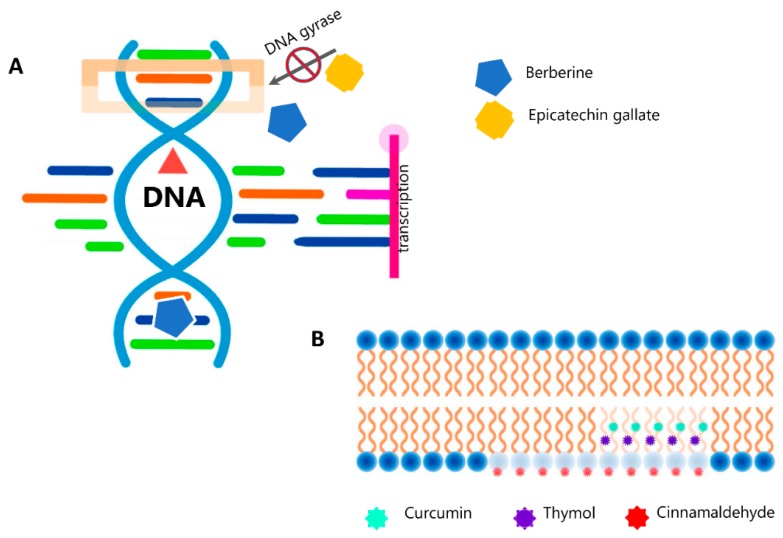
Antimicrobial action of secondary metabolites of plants (SMoPs). (**A**). Effect of SMoPs on the DNA replication and transcription. Epicatechin gallate inhibits bacterial DNA gyrase by binding to the ATP binding site of the gyrase B subunit [103]. Berberine inhibits DNA synthesis by affecting the activity of DNA topoisomerase [93]. An intercalative mode of binding for this alkaloid to DNA was also suggested. [104]. (**B**). Curcumin [57] as well as cinnamaldehyde [3] penetrate to membrane bilayer and enhance its permeability both in Gram-positive (*S. aureus*) and Gram-negative (*E. coli*) bacteria. Disruption of membrane integrity is the major mechanism of action of thymol against *S. typhimurium* [105].

**Figure 3 antibiotics-09-00170-f003:**
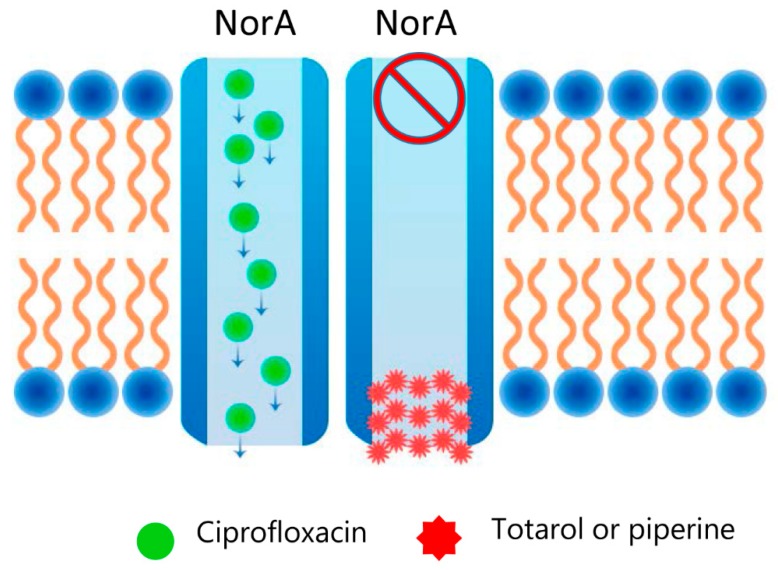
Plugging the efflux pumps with SMoPs. Piperine, the major plant alkaloid present in black pepper (*Piper nigrum*) and long pepper (*Piper longum*), or totarol—a diterpene from *Podocarpus totara*, inhibits NorA-mediated ciprofloxacin efflux from *S. aureus* cells [112,113,114].

**Table 1 antibiotics-09-00170-t001:** Plant compounds demonstrating antimicrobial activity against important human pathogens.

Pathogen	Substance	Group	Plant Source	MIC*, μg/mL	Mechanism	Ref
*Acinetobacter baumannii*	allicin	organosulfur compound	*Allium sativum*	16	DNA and protein synthesis inhibitor	[51]
*Pseudomonas aeruginosa*	conessine	alkaloid	*Holarrhena floribunda, Holarrhena antidysenterica, Funtumia elastica*	40	efflux pump inhibitor	[52]
	allicin	organosulfur compound	*Allium spp.*	64	DNA and protein synthesis inhibitor	[51]
	thymol	terpenoids	*Thymus vulgaris, Thymus capitatus*	5	cell membrane disturbance	[53]
	carvacrol			7	disintegration of the outer membrane	[53]
	eugenol		*Syzygium aromaticum and Eugenia caryophillis*	150–300		[54]
*Escherichia coli*	berberine	alkaloid	*Berberis vulgaris*	4 mM	inhibition of the cell division protein FtsZ	[55]
	p-OH-benzoic acid	benzoic acid	*Scrophularia spp.*	>2000		[56]
	curcumin	diarylheptanoid	*Curcuma longa*	25–100	damaging of bacterial membrane	[57]
	apigenin	flavonoids	*Matricaria chamomilla*	200		[58]
	quercetin		*Capparis spinosa*	300	efflux pump inhibitor	[58]
	epigallocatechin gallate		*Camellia sinensis*	200 μM		[59]
	(+)-Catechin hydrate		*Camellia sinensis*	>2000		[56]
	genistein		*Glycine max*	>2000		[56]
	protocatechuic acid	phenolic acids	*Scrophularia frutescens*	>2000		[56]
	gallic acid		*Vitis rotundifolia*	>2000		[56]
	hydroquinone	phenol	*Vaccinium myrtillus*	>2000		[56]
	resveratrol	polyphenol	*Vitis vinifera*	1300		[56]
	eugenol	terpenoids	*Syzygium aromaticum and Eugenia caryophillis*	>2000		[56]
	thymol		*Thymus capitatus; Tyhmus vulgaris*	8; 800	cell membrane disturbance	[53,56]
	carvacrol		*Thymus capitatus; Tyhmus vulgaris*	8; 100	disintegration of the outer membrane	[53,56]
*Klebsiella pneumoniae*	osthole	coumarin	*Cnidium monnieri*	125	DNA gyrase inhibitor	[60]
	allicin	organosulfur compound	*Allium sativum*	128	DNA and protein synthesis inhibitor	[51]
*Enterococcus faecalis*	taxifolin	flavonoids	*Pinus roxburghii*	128		[61]
	eriodictyol		*Eriodictyon californicum*	256		[61]
	naringenin		*Citrus paradisi*	256		[61]
*Staphylococcus aureus (including MRSA)*	piperine	alkaloid	*Piper nigrum*	100	efflux pump inhibitor	[62]
	aegelinol	coumarins	*Ferulago campestris*	16		[63]
	agasyllin		*Ferulago campestris*	32		[63]
	osthole		*Cnidium monnieri, Angelica archangelica and Angelica pubescens*	125	DNA gyrase inhibitor	[60]
	sophoraflavanone B	flavonoids	*Desmodium caudatum*	15.6–31.25	direct interaction with peptidoglycan	[64]
	genistein		*Glycine max*	100 μM	efflux pump inhibitor	[65]
	chrysoplentin		*Artemisia absinthemum*	6.25	efflux pump inhibitors	[66]
	quercetin		*Capparis spinosa*	75		[58]
	kaempferol		*Moringa oleifera, Sambucus nigra, Aloe vera*	125		[67]
	apigenin, kaempferol, rhamnetin, quercetin, myricetin		in many plants	>150		[68]
	luteolin		*Reseda luteola*	75		[68]
	allicin	organosulfur compounds	*Allium sativum*	32, 64	DNA and protein synthesis inhibitor	[51]
	farnesol	terpenes	*Vachellia farnesiana*	20 (MBC)	cell membrane disturbance	[69]
	nerolidol		*Cannabis sativa*	40 (MBC)		[69]
	thymol	terpenoids	*Thymus capitatus*	6.5	cell membrane disturbance	[53,70]
	carvacrol		*Thymus capitatus*	7	disintegration of the outer membrane	[53]
	plumbagin	naphthoquinone	*Plumbago zeylanica*	4–8		[71]
*Helicobacter pylori*	aegelinol, agasyllin	coumarins	*Aegle marmelos, Ferulago asparagifolia Boiss*	5–25	DNA gyrase inhibitor	[63]
	cinnamaldehyde	flavonoids	*Cinnamomum spp.*	2	cell membrane disturbance	[72]
	quercetin		*Polymnia fruticosa*	330.9 μM	inhibit some enzymes involved in the type II fatty acid biosynthesis pathway (FabZ)	[73]
	apigenin		*Polymnia fruticosa*	92.5 μM		[73]
	sakuranetin		*Polymnia fruticosa*	87.3 μM		[73]
	apigenin		*Matricaria chamomilla, Apium graveolens, Apium graveolens*	25	efflux pump inhibitors	[58]
	quercetin		*Capparis spinosa*	100–200		[58]
	iberin, erysolin	organosulfur compounds	*Iberis spp., Erysimum spp.*	32 (MIC_90_)		[74]
	cheirolin, berteroin, alyssin		*Cheiranthus cheiri, Berteroa incana, Alyssum sp.*	16 (MIC_90_)		[74]
	hirsutin		*Rorippa sp., Nasturnium officinale*	8 (MIC_90_)		[74]
	eugenol	terpenoid	*Syzygium aromaticum and Eugenia caryophillis*	2	cell membrane disturbance	[72]
	juglone derivatives	naphthoquinones	*Reynoutria japonica*	0.06–6.3 μM		[75]
*Campylobacter* *spp.*	resveratrol	polyphenol	*Vitis vinifera*	313		[76]
*Salmonella typhii*	agasyllin	pyranocoumarin	*Ferulago campestris*	32	DNA gyrase inhibitor	[63]
	aegelinol	pyranocoumarin	*Aegle marmelos, Ferulago asparagifolia Boiss*	16–32		[63]
*Streptococcus pneumoniae*	allicin	organosulfur compound	*Allium sativum*	32, 64	DNA and protein synthesis inhibitor	[51]
*Mycobacterium tuberculosis*	evocarpine, evodiamine	alkaloids	*Evodiae fructus*	5–20 10–80	inhibition of ATP-dependent MurE ligase of *Mycobacterium tuberculosis*, an enzyme required for the biosynthesis of peptidoglycan	[77]
	piperine	alkaloid	*Piper nigrum*	50–100	efflux pump inhibitor	[78]
	andrographolide	diterpenoid	*Andrographis paniculata*	250	probable target for andrographolide is aminoglycoside 2′-N-acetyltransferase	[79]
	ent-kaurane, kaurane, grayanane	diterpenoids	*Croton tonkinensis*	<12.5		[80]
	plumericin; iso-plumericin	iridoid lactone	*Plumeria bicolor*	1.5–2.1; 2.0–2.6		[81]
	artemisinin (synthetic analogs)	sesquiterpene lactone	*Artemisia annua*	>25.0		[82]

* MIC, minimal inhibitory concentration, is given in µg/mL, otherwise specified. MBC, minimal bactericidal concentration.

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
