# Peer review of "Plant Secondary Metabolites in the Battle of Drugs and Drug-Resistant Bacteria: New Heroes or Worse Clones of Antibiotics?"

_antibiotics, 2020, doi:10.3390/antibiotics9040170_

Round 1
Reviewer 1 Report
In connection with the spread of antibiotic-resistant pathogens, an important task is the search for new effective drugs. The review is devoted to the analysis of the described biologically active substances of plant origin (secondary metabolites of plants - SMoPs), namely those that have antimicrobial properties. The authors of the review are looking for the answer to the question: do antibacterial agents derived from plants have a chance to become a panacea against infectious diseases in the "post-antibiotics era?"SMoPs with antibiotic properties, their classification, mechanisms of action, targets in bacterial cells, toxicity, the possibility of use as drugs, efficacy against resistant forms of bacteria are examined. It is noted that, as with conventional antibiotics, bacterial forms resistant to SMoPs can be produced. The authors conclude that combining SMoPs with antibiotics results in improved pharmacological activity and can minimize toxicity.This review does not include studies of new antimicrobial compounds produced by plant symbionts. A striking example of such compounds is the antitumor antibiotic taxol. However, this may be the subject of a special review. Also, the comments could include the absence of publications by J. Berdy (2005 and 2012), these are reviews on this topic.
This review is useful for specialists in this field and can be published.
Author Response
- Also, the comments could include the absence of publications by J. Berdy (2005 and 2012), these are reviews on this topic.
Thank you for this comment. Both reviews of János Bérdy (J. Bérdy. Bioactive microbial metabolites. J. Antibiot. 58(1): 1–26, 2005; and J. Bérdy. Thoughts and facts about antibiotics: where we are now and where we are heading. J. Antibiot. 2012, 65, 385–395) are devoted mainly to biologically active metabolites of microbial origin, while the main topic of our article is secondary metabolites produced by plants; so they were omitted from our citation list. Nevertheless, the review of this author of 2012 contains discussion about searching for new natural compounds that echo our own discussion on this subject in our manuscript, therefore I included this review into our citation list (line 97, ref. 37).
Reviewer 2 Report
In my opinion the topic of the review is very interesting and hot as bacterial resistance to antibiotics increases. I have some questions and comments:
- In section 3 “Antimicrobial properties of SMoPs: the reserve players against the life-threatening pathogens” authors should add information about an acronym (ESCAPE) encompassing the names of 6 bacterial pathogenes commonly associated with antimicrobial resistance (E. faecium, S. aureus, C. difficile, A. baumanni, P. aeruginosa and Enterobacteriaceae).
- 2. From table 2, the authors should remove information concerning secondary metabolites whose MIC is greater than 256 ug / ml (eg benzoic acid, phenol). Table 2 should be supplemented with naphthoquinones that have strong bactericidal activity.
- 3. In section 5.3. authors should emphasize biofilm resistance to antibiotics and secondary metabolites and the difficulty of fighting it.
Author Response
- In section 3 “Antimicrobial properties of SMoPs: the reserve players against the life-threatening pathogens” authors should add information about an acronym (ESCAPE) encompassing the names of 6 bacterial pathogenes commonly associated with antimicrobial resistance (E. faecium, S. aureus, C. difficile, A. baumanni, P. aeruginosa and Enterobacteriaceae).
We added the information about ESKAPE pathogens and referred to the article in which this acronym was firstly mentioned (lines 149-152, “These pathogens were determined by WHO in 2017 as the most life-threatening bacteria due to their rapidly developing resistance to drugs, this list includes among others those pathogens that WHO has assigned a critical priority I and II - the so-called ESKAPE group (an acronym for Enterococcus faecium, Staphylococcus aureus, Klebsiella pneumoniae, Acinetobacter baumannii, Pseudomonas aeruginosa and Enterobacter spp.) [83]”).
- From table 2, the authors should remove information concerning secondary metabolites whose MIC is greater than 256 ug / ml (eg benzoic acid, phenol). Table 2 should be supplemented with naphthoquinones that have strong bactericidal activity.
This comment is not quite clear. In our article, there is only one Table 2, namely in the Appendix (Table S2). But it only lists MICs for antibiotics, not for the secondary plant metabolites. We guess that the reviewer may have been referring to Table 1 (in the text) and probably meant the data from the work of Guttierez (Gutiérrez et al., 2017, ref. 56) where MICs defined for a number of SMoPs are unusually high (> 2000 µg/ml). However, we would like to emphasize that the purpose of Table 1 was just to summarize up the data from different studies, not to find the best antimicrobial agent. In addition, the minimum inhibitory concentrations for secondary plant metabolites are generally higher than those for current antibiotics of microbial origin, and we pointed this out in the text (lines 184-187). Therefore, we think all the data of Table 1 should be kept.
As to the second recommendation, we have inserted some latest examples of the bactericidal activity of naphthoquinones plumbagin and juglone derivatives in the Table 1, with correspondent references (see Table 1).
- In section 5.3. authors should emphasize biofilm resistance to antibiotics and secondary metabolites and the difficulty of fighting it.
The clarifying sentence “Due to their high tolerance to antibiotics, biofilms that are typically the cause of chronic, nosocomial, and medical device-related infections are the great problem in the clinic.” was inserted in the text (lines 317-319).
- Additionally, the list of references was re-formatted, because of the several new references were introduced (in accordance with the comments above).